# Detection of Early Endothelial Dysfunction by Optoacoustic Tomography

**DOI:** 10.3390/ijms24108627

**Published:** 2023-05-11

**Authors:** Carsten Höltke, Leonie Enders, Miriam Stölting, Christiane Geyer, Max Masthoff, Michael T. Kuhlmann, Moritz Wildgruber, Anne Helfen

**Affiliations:** 1Clinic for Radiology, University Hospital Münster, 48149 Münster, Germany; leonie.enders23@gmail.com (L.E.); miriam.stoelting@ukmuenster.de (M.S.); christiane.geyer@uni-muenster.de (C.G.); max.masthoff@ukmuenster.de (M.M.); moritz.wildgruber@med.uni-muenchen.de (M.W.); anne.helfen@ukmuenster.de (A.H.); 2European Institute for Molecular Imaging, WWU Münster, 48149 Münster, Germany; kuhlmam@uni-muenster.de; 3Department of Radiology, University Hospital, LMU Munich, 81377 Munich, Germany

**Keywords:** endothelial dysfunction, optoacoustic tomography, atherosclerosis, altered shear stress, integrins, RGD-mimetics

## Abstract

Variations in vascular wall shear stress are often presumed to result in the formation of atherosclerotic lesions at specific arterial regions, where continuous laminar flow is disturbed. The influences of altered blood flow dynamics and oscillations on the integrity of endothelial cells and the endothelial layer have been extensively studied in vitro and in vivo. Under pathological conditions, the Arg-Gly-Asp (RGD) motif binding integrin α_v_β_3_ has been identified as a relevant target, as it induces endothelial cell activation. Animal models for in vivo imaging of endothelial dysfunction (ED) mainly rely on genetically modified knockout models that develop endothelial damage and atherosclerotic plaques upon hypercholesterolemia (ApoE^−/−^ and LDLR^−/−^), thereby depicting late-stage pathophysiology. The visualization of early ED, however, remains a challenge. Therefore, a carotid artery cuff model of low and oscillating shear stress was applied in CD-1 wild-type mice, which should be able to show the effects of altered shear stress on a healthy endothelium, thus revealing alterations in early ED. Multispectral optoacoustic tomography (MSOT) was assessed as a non-invasive and highly sensitive imaging technique for the detection of an intravenously injected RGD-mimetic fluorescent probe in a longitudinal (2–12 weeks) study after surgical cuff intervention of the right common carotid artery (RCCA). Images were analyzed concerning the signal distribution upstream and downstream of the implanted cuff, as well as on the contralateral side as a control. Subsequent histological analysis was applied to delineate the distribution of relevant factors within the carotid vessel walls. Analysis revealed a significantly enhanced fluorescent signal intensity in the RCCA upstream of the cuff compared to the contralateral healthy side and the downstream region at all time points post-surgery. The most obvious differences were recorded at 6 and 8 weeks after implantation. Immunohistochemistry revealed a high degree of α_v_-positivity in this region of the RCCA, but not in the left common carotid artery (LCCA) or downstream of the cuff. In addition, macrophages could be detected by CD68 immunohistochemistry in the RCCA, showing ongoing inflammatory processes. In conclusion, MSOT is capable of delineating alterations in endothelial cell integrity in vivo in the applied model of early ED, where an elevated expression of integrin α_v_β_3_ was detected within vascular structures.

## 1. Introduction

Atherosclerosis is an inflammatory disorder of the endothelial cell (EC) layer originating from early endothelial dysfunction (ED) [1]. Atherosclerotic lesions predominantly develop at regions where vessel branches or curvatures are present and the regular blood flow is disturbed. The influence of altered blood flow dynamics and oscillations on the integrity of ECs and the endothelial layer leads to EC activation and the manifestation of a pro-inflammatory “atheroprone” phenotype [2,3]. Essential characteristics of ED are a reduction in nitric oxide (NO) synthesis and the marked upregulation of reactive oxygen species (ROS) in ECs, but biomarkers of early ED that can be utilized for clinical diagnosis are missing [4]. In the vasculature, integrins play an essential role for the integrity of the endothelial cell basement membrane–vascular smooth muscle cell layer system [5,6,7]. Integrins are a large family of heterodimeric transmembrane proteins mediating cell–cell and cell–extracellular matrix (ECM) interactions. In mammals, there are 18 α- and 8 β-subunits that combine to form 24 different integrin subtypes [5,8]. Among them are collagen- and laminin-binding integrins, such as α_2_β_1_ or α_6_β_4_, which mediate the firm attachment of ECs to the basement membrane in intact and quiescent blood vessels. In activated endothelium or during inflammation, α_v_β_3_ and α_5_β_1_ integrins, which bind the RGD sequence in fibronectin and vitronectin, are the most abundant [9]. In response to altered shear stress, these integrins are able to act as mechanotransducers by transmitting mechanically initiated signals into the cell, thereby activating intracellular signaling pathways that may stimulate inflammation [6,10]. Integrin α_v_β_3_ has been shown to contribute to neointima formation, angiogenic processes, and smooth muscle cell (SMC) proliferation in a hamster atherosclerosis model [11]. In addition, RGD-mediated integrin α_5_ signaling has been identified as an essential element of vascular remodeling processes in a model of hypertension [12]. Additionally, in 2005, Cheng and colleagues developed a perivascular shear stress modifier (originally referred to as a “cast” and later termed a “cuff”) with an inner cone shape that produces reduced shear stress upstream, increased shear stress within, and oscillatory blood flow downstream of the cuff, when fixed around a vessel such as the carotid artery [13,14]. This cuff model has frequently been used to examine the influence of altered blood flow dynamics and oscillations on the integrity of ECs and the development of atherosclerotic lesions in apolipoprotein E deficient (ApoE^−/−^) mice. These preclinical in vivo investigations utilize positron emission tomography (PET) to visualize alterations in target expression [15,16]. MRI has been used to evaluate the effect of high-fat diet on the coronary vasculature in a model of early ED in wild-type mice [17]. While PET and MRI suffer from radiation issues and rather cost-intensive instrumentation, we aimed to investigate the capability of multispectral optoacoustic tomography (MSOT) to delineate early responses to altered shear stress in carotid arteries after cuff implantation in wild-type mice. MSOT, as a hybrid optical/ultrasound technique, combines the high sensitivity of laser-induced optical fluorophore excitation with the advantageous tissue-penetrating characteristics of ultrasound waves, which enables the delineation of anatomical structures and pathophysiological processes with high spatial resolution in real time. We utilized an exogenous RGD-mimetic fluorescent probe to detect enhanced integrin α_v_β_3_ expression as an early sign of endothelial impairment before the onset of severe vascular changes.

## 2. Results

### 2.1. MSOT Imaging

In contrast to ApoE^−/−^ mice [14,16,18], the implantation of a tapered cuff around the carotid arteries of wild-type CD-1 mice does not result in plaque formation upstream or downstream of the cuff. A macroscopic plaque deposition within the RCCA of ligated animals was not observed throughout the time course of the experiment, up to 12 weeks (Figure 1A). We started MSOT imaging experiments two weeks after surgery, when inflammatory processes due to ligation should have subsided. Subsequently, we applied MSOT every 2 weeks until 12 weeks after surgery. MSOT imaging yielded data for total hemoglobin distribution (HbT) within the observed region, as depicted in Figure 1B. The highest signal originated from the adjacent jugular veins, framing the carotid arteries. In contrast to the manipulated right vessel, the LCCA typically showed a more uniform signal distribution of HbT, emphasizing the disturbed flow conditions around the cuff. In order to visualize signs of early endothelial activation due to modified shear stress in treated mice in vivo, we injected a small molecular fluorescent integrin probe, which binds α_v_β_3_ with high affinity [19]. When analyzing the signal distribution of the injected probe, a more detailed picture was observed (Figure 1C). The most pronounced signal originated from the cuff itself (Figure 1C) which lit up markedly in the chosen spectral region due to material characteristics. It generated a defined but low signal in the IRDye800 spectral region, as observed in a phantom experiment (see Appendix A). Therefore, we omitted the cuff when drawing regions of interest. Data from the upstream region of the RCCA were compared to data from the downstream region and the unaffected LCCA. At all time points post-surgery (2, 4, 6, 8, 10, and 12 weeks, Figure 1D) the mean signal intensity from the upstream region was significantly higher (25 ± 7–32 ± 10 au) than the signals from the downstream region (18 ± 6–21 ± 9 au) and the LCCA (17 ± 4–22 ± 7 au). The highest signals were recorded 6 weeks after surgery, and the most significant differences 8 weeks after surgery.

### 2.2. CIMT Evaluation

In the following, we investigated explanted carotid artery specimens for signs of endothelial damage and development of atherosclerotic plaques. Elastica von Gieson staining was routinely performed on all specimens (Figure 2). In some, but not all, samples of mice 12 weeks after surgery, a considerable intima thickening, including endothelial impairment, was observed (Figure 2A), which in some cases extended into the intra-cuff region (Figure 2B). At earlier time points after surgery, these effects were not observed (Figure 2C). The contralateral vessels were unaffected by the procedure at all time points and did not develop any signs of endothelial impairment (Figure 2D).

In a number of samples, the carotid intima–media thickness (CIMT) was measured, since CIMT is often used as a surrogate marker of subclinical and early atherosclerosis [20]. We compared early (2 and 4 weeks post-surgery) and late (10 and 12 weeks post-surgery) explants of carotid arteries upstream of the implant and used left carotid arteries (LCCA) as control (Figure 2E). While the left carotid artery showed unchanged morphology at early and late time points, with CIMT values from 10.8 to 19.1 µm (means: 16.1 ± 3.7 and 15.5 ± 1.2, resp., *n* = 4 each), a significantly enhanced thickness was observed in the late tissue samples, where values between 24.7 and 69.2 µm were measured (mean: 40.3 ± 15.6, *n* = 6). Values for the early RCCA specimen ranged from 10.0 to 38.8 µm (mean: 23.1 ± 11.2, *n* = 9).

### 2.3. Immunohistochemistry

For the verification of enhanced integrin α_v_β_3_ expression, we investigated ligated carotid arteries by immunohistochemistry upstream and downstream of the cuff. For control, the non-ligated LCCA was used. Figure 3 shows a comparison of tissue from a mouse eight weeks post-surgery, where differences in in vivo signal intensities were highly significant. Staining for the combination of α_v_β_3_ (CD51/CD61, Figure 3A) and for the single subunits (alpha-V, Figure 3B, and beta-3, Figure 3C) were carried out. The left carotid artery (LCCA, control) shows signal intensity for α_v_β_3_ and alpha-V, but only slightly for beta-3, while the upstream region of the ligated vessel shows a strong signal intensity for α_v_β_3_ and alpha-V, and also a distinct signal intensity for beta-3. The downstream region only exhibits a weak signal intensity in all stainings. Integrin beta-3 seems to be completely absent and α_v_β_3_ signal intensity is lower than in control tissue, while the alpha-V signal is comparable to LCCA signal intensity. Additional staining for macrophages with CD68 antibody reveals early immune cell infiltration. Enhanced signal intensity in the tissue surrounding the vessels (Figure 4A) was recorded in RCCA specimens. Already, at early time points (4 weeks post-surgery), an infiltration can be observed, also showing positive cells within the vessel wall (Figure 4B), as well as within the scar tissue surrounding the cuff itself (Figure 4C).

## 3. Discussion

In the present study we wanted to elucidate the applicability of the MSOT hand-held optoacoustic imaging probe in combination with an integrin α_v_β_3_-targeted fluorescent small molecular RGD mimetic for the detection of early endothelial dysfunction in a cuff model of altered shear stress in the carotid arteries of wild-type mice. Multispectral optoacoustic tomography (MSOT) is based on the detection of ultrasonic waves generated by the thermoelastic expansion of tissue after pulsed laser excitation. The handheld probe we applied was an advanced preclinical version of the one described by Buehler et al., which was originally developed for clinical translation [21,22,23]. The effects of altered shear stress within arterial vessels utilizing this cuff model have most often been described by investigating plaque development in ApoE^−/−^ mice receiving a high-fat diet [16,24,25]. Bertrand and colleagues used a model of balloon injury in the abdominal aorta of high-cholesterol diet-fed rabbits. They were able to visualize an upregulation of ICAM-1 and collagen I fragments using a combination of intravascular ultrasound and near infrared fluorescence imaging [26]. Additionally, a combined fluorescence and magnetic resonance imaging approach has been described, identifying integrin α_v_β_3_ on activated macrophages of carotid arteries in a murine ApoE^−/−^ model [27]. However, all of the described studies use models of substantial physiological deficiencies (genetically modified animals plus western diet or massive intravascular interventions). To our knowledge, a study of endothelial activation in wild-type animals without additional high-fat diet utilizing the cuff model has not yet been described. By applying an endogenous contrast agent targeted at integrin α_v_β_3_, the aforementioned preferential activation of regions upstream of the constriction can also be observed in this model. The involvement of integrins in endothelial responses to hemodynamic forces, including altered shear stress, was first described in the 1990s [28,29]. Thereafter, integrins were described as mechanotransducers, translating an extracellular mechanical stimulus into cells [30,31,32]. The sensing and transduction of vascular force perturbations, however, is not yet understood in its entirety. A variety of endothelial cell surface components can act as mechanotransducers, including ion channels, chemokine receptors, and the glycocalyx [33]. Integrins accomplish mechanotransduction via the recruitment of cytoskeletal proteins, such as talin and vinculin, inside cells that connect integrins to actin filaments, and are the basis of focal adhesions [32,34]. The resulting signaling leads to the phosphorylation and activation of non-receptor tyrosine kinases (e.g., FAK and c-Src), which in turn activate Ras family GTPases. Active Ras subsequently triggers the activation of mitogen-activated protein kinase (MAPK). Ultimately, these signaling pathways lead to the phosphorylation of transcription factors such as nuclear factor-kappa B (NF-κB), and, thereby, to the activation or suppression of genes responsible for the adaptation of cellular function and morphology [2,34]. One consequence of these signaling and activation processes is the prompting of dynamic changes in the composition of the endothelial basement membrane from laminin- and collagen IV-rich to fibronectin-rich ECM [6]. In turn, the integrin expression pattern of endothelial and vascular smooth muscle cells changes from laminin- and collagen IV-binding (α_2_β_1_ or α_6_β_4_) to fibronectin-binding (α_v_β_3_ or α_5_β_1_) [5]. Our results suggest an upregulation of α_v_β_3_ in the upstream region of the cuff compared to the downstream region and the control LCCA, even in wild-type animals without a high-fat diet and at an early time point, prior to plaque formation. This elevated expression of integrin α_v_β_3_ could be confirmed by immunohistochemistry in the upstream regions in tissue from animals 12 weeks post-surgery. However, in future studies, the contribution of other integrins, such as α_v_β_5_ and α_5_β_1_, which also bind the RGD motif and are described to contribute to endothelial cell inflammation, needs to be investigated [35,36,37]. Additionally, immunohistochemistry shows that in tissue within the cuff and surrounding the vessel, from late as well as from early time points, an enhanced population of macrophages is present, which hints at ongoing inflammatory processes. Thus, shear-stress-induced early endothelial activation with inflammatory potential can be concluded. The present study shows that the underlying processes can be visualized by MSOT in combination with an endogenous fluorescent probe. The excellent spatial resolution and high sensitivity makes this technique superior to common optical imaging methods. As an additional benefit, it offers anatomical information from the hybrid ultrasound, which is implemented in the 3D imaging setup. A distinct drawback in the present study was the marked signal from the cuff itself, which was attributed to material characteristics. In future studies, the utilization of different models for the induction of modified shear stress is warranted. The MSOT technique has already been translated to patient examinations and a commercially available clinical MSOT scanner is currently evaluated for a number of diseases, including late-stage carotid atherosclerosis [23,38]. Approaches utilizing exogenous contrast agents, however, must be conducted with caution, since approved agents apart from indocyanine green (ICG) and methylene blue are not available.

## 4. Methods

### 4.1. Animals

In total, 60 7-week-old wild-type CD-1 mice were either purchased from Charles River Laboratories (Sulzfeld, Germany) or received from the central animal facility of the Medical Faculty of the University of Münster and kept in groups of up to 5 animals in pathogen-free Makrolon cages with filter cap and provided with enrichment material, stored at constant temperature (24 ± 2 °C) and relative humidity (46 ± 4%) in a rodent flow cabinet. Animals had free access to water and food (normal chow diet, 1324 Best, Altromin, Lage, Germany). After one week of acclimation, surgery was performed in 2–3 consecutive days. A tapered cuff was implanted around the right common carotid artery (RCCA), which constricts the vessel diameter, and thereby modifies the flow properties and endothelial shear stress in a defined manner (Figure 5A–C). Briefly, after anesthesia and shaving of the coronal chest and neck area, the common carotid arteries could be accessed through a 4–5 mm medial incision starting from the top of the sternum. The right common carotid artery was dissected free from the surrounding connective tissue and separated from the vagus nerve. After placing the thread of silk suture (6–0) around the vessel, the cuff was fixed around the right common carotid artery by tightening the circumferential silk loops. The entry wound was closed by a small amount of 6–0 prolene suture, and the animals were returned to their cages for recovery. The surgical procedure has been described in detail elsewhere [39]. Surgery was performed under combined inhalation anesthesia (2% isoflurane, 0.5 l O_2_/min) and intraperitoneally injected narcotics (0.04 mg Fentanyl, 4 mg Midazolam/kg bodyweight) for effective pain treatment during the intervention. For the treatment of any post-surgical pain Carprofen (5 mg/kg bodyweight) was administered subcutaneously at the end of the surgery and again, if necessary, the following days. To detect any health issues, animals were checked on a daily basis. All animal experiments were performed in accordance with the legal requirements of the German Animal Welfare Law (TierSchG, TierSchVersV) and were approved by the local authorizing agency (State Office for Nature, Environment and Consumer Protection North Rhine-Westphalia (Landesamt für Natur, Umwelt und Verbraucherschutz, LANUV); protocol no. 84–02.04.2016.A511).

### 4.2. Fluorescent Probe

A small molecular RGD-mimetic compound previously developed was labeled with the NHS ester of IRDye800cw (LI-COR Biosciences, Bad Homburg, Germany) to meet the requirements of the MSOT system [19,40]. The compound was purified with HPLC (Shimadzu Prominence gradient RP-HPLC system with diode array detection, Shimadzu Deutschland GmbH, Duisburg, Germany), acetonitrile, and purified water containing 0.1% trifluoro acetic acid as mobile phases, and Nucleosil^®^ 100-5 C18 RP columns. Mass spectrometry was performed using an Orbitrap LTQ XL (Thermo Scientific, Dreieich, Germany) spectrometer with nanospray capillary inlets: *m*/*z* = 1841.51266 [M-3H+2Na]^−^; calc.: C_81_H_98_N_10_O_23_S_5_F_3_Na_2_^−^ = 1841.51628. The concentration of the reconstituted (purified water) probe solutions were measured by determining the absorption of serially diluted aliquots and applying Lambert–Beer’s law, using the excitation coefficient of IRDye800cw dye provided in the literature. Concentrations between 360 µM and 890 µM were used.

### 4.3. Multispectral Optoacoustic Tomography Imaging

Animals were divided into 6 groups for imaging. Groups of 8–10 animals were imaged at the following time points after surgery: 2, 4, 6, 8, 10, and 12 weeks (some mice suffered premature death before the start of imaging procedures). For imaging, animals were prepared by shaving the upper chest region on day 1, followed by injection of the probe (2.0 nmol per animal) the next day, and MSOT imaging 3 h after injection. The handheld cup-shaped probe (Figure 6) used in combination with the MSOT inVision512 system (iThera Medical, Munich, Germany) had the following specifications: Probe dimensions (L × W × H): 130 × 85 × 85 mm; Field of view: 15 mm × 15 mm (lateral × depth); Resolution for depth of 10 mm (±1.5 mm) in the lateral image center: in-plane (xy) < 160 μm, out-of-plane (z) 160 μm; Center frequency: 8.0 MHz. Animals were placed on the dedicated animal bed with isoflurane anesthesia and gentle heating. The upper chest and throat region were covered with gel for ultrasound imaging. Images were acquired at nine wavelengths from 680nm to 950nm, in triplicate, and processed by the viewMSOT 4.0 software. Images were reconstructed using the manufacturer’s back projection algorithm. Spectral unmixing was performed for oxygenated and deoxygenated hemoglobin (HbO_2_, Hb) and for IRDye800 by linear regression methods.

### 4.4. Multispectral Optoacoustic Tomography Data Analysis

Data were analyzed by processing the maximum intensity projections (MIPs) of the target region. Background data were recorded at a wavelength of 850 nm and used for the localization of the carotid arteries. Background data and frames containing IRDye800 fluorescence intensity with fixed scales were separately transferred to ImageJ. Regions of interest were drawn around the upstream and downstream regions of the RCCA and around the corresponding LCCA in the synchronized channel windows. The cuff itself was omitted. Intensity data, expressed in arbitrary units (au), were transferred to Microsoft Excel, and then further analyzed and visualized by GraphPad Prism 7.05 (GraphPad Software, La Jolla, CA, USA) using one-way ANOVA and Holm–Sidak’s multiple comparison test. Significant differences were concluded from *p*-values < 0.05. Data are presented by box plots showing min-to-max whiskers. Due to data inconsistencies and focusing issues, not all data sets could be analyzed accordingly, resulting in partially reduced animal numbers.

### 4.5. Immunohistochemistry

For histological analysis, the carotid arteries were harvested after mice were euthanized under deep anesthesia. The cuff half shells were carefully opened and residual connective tissue and fat was removed. Tissue samples from the dissected arteries were then fixated in 4% neutral buffered formalin for 24 h and subsequently embedded in paraffin according to standard protocols. Sections of 5 µm thickness were then manufactured using a microtome (RM2235, Leica; Wetzlar, Germany). Antigen retrieval was performed using the Vector 3300 unmasking solution. Sections were stained for integrin alpha-v (ABIN3022888), integrin beta-3 (ABIN6758960), and CD51/61 heterodimer (ABIN674784) using a goat anti-rabbit AlexaFluor647 fluorescent secondary antibody (Jackson) and DAPI counterstain for visualization. Further stainings included CD68 (ab125212, Abcam, Cambridge, UK), smooth muscle actin (SMA, F3777, Sigma, Kawasaki City, Japan) and Elastica van Gieson (Morphisto, Offenbach am Main, Germany), and were performed according to the manufacturer’s protocol. Negative control experiments were performed without first antibody. Every staining was performed on at least two samples of each time point. Sections were imaged using a Nikon Eclipse 50i microscope and documented using NIS-Elements Br 3.22 software (Nikon Corporation, Tokyo, Japan).

### 4.6. Carotid Intima–Media Thickness (CIMT)

Elastica van Gieson stainings were utilized to determine the intima media thickness of the carotid arteries. Left arteries (LCCA) served as controls, and the upstream region of the RCCA was the main target region. We compared early (2–4 weeks post-surgery) and late (10–12 weeks post-surgery) time points after cuff implantation and measured three different, randomly selected regions of each section. Distances were evaluated using the measurement tool from NIS-Elements Br 3.22 software (Nikon Corporation, Tokyo, Japan) at 40× magnification. Data are presented as mean ± sem; statistical significance was assessed by one-way ANOVA and Holm–Sidak’s multiple comparison test using GraphPad Prism 7.05 software as described above.

## 5. Conclusions

Modified shear stress in the common carotid artery, caused by an implanted tapered cuff, results in enhanced integrin expression upstream of the ligation, which can be identified by a small molecular fluorescent α_v_β_3_-targeted probe in combination with a hand-held multispectral optoacoustic tomography (MSOT) system. The observed integrin upregulation is putatively a sign of early endothelial activation and dysfunction. The potential of MSOT to non-invasively capture the associated processes is demonstrated and emphasizes the high spatial resolution of this technique.

## Figures and Tables

**Figure 1 ijms-24-08627-f001:**
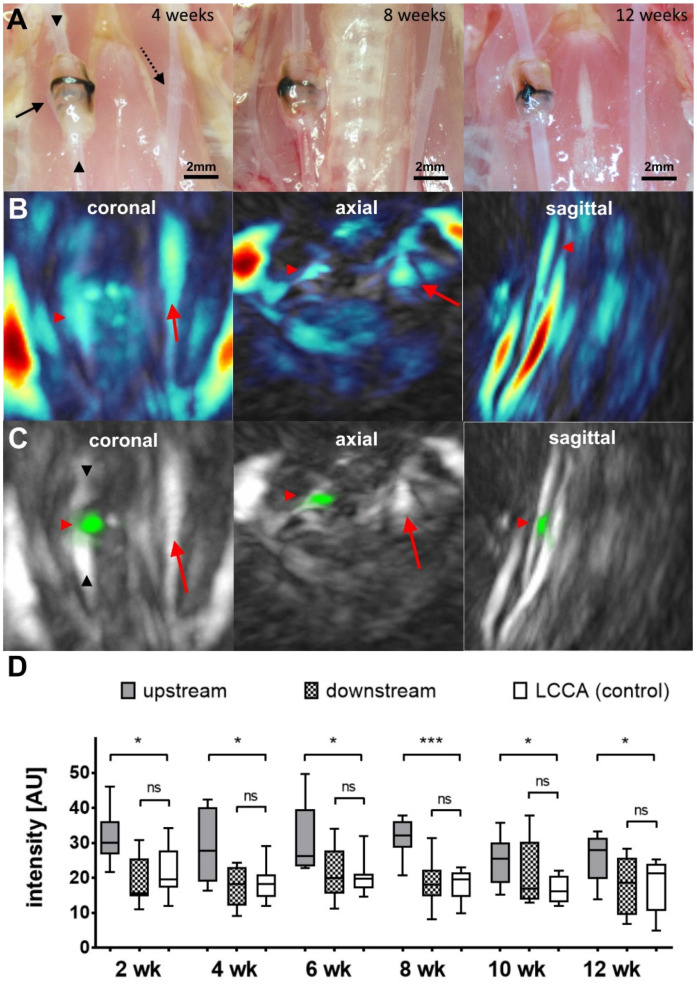
(**A**) Intraoperative images of the implanted cuff regions (black arrow) at 4, 8, and 12 weeks post-implantation, emphasizing the upstream (upward arrowhead) and downstream (downward arrowhead) parts of the RCCA, as well as the LCCA (dotted arrow). The complete absence of plaque material deposition can clearly be observed at all time points. (**B**) MSOT images showing the exemplary distribution of total hemoglobin (HbT) in vivo in one mouse 8 weeks post-surgery. The highest signal originates from the jugular veins in all three projections. The location of the cuff site is indicated by a red arrowhead. The LCCA is marked by a red arrow. In the sagittal view, signals from the carotid arteries interfere with each other. (**C**) Maximum intensity projections (MIPs) of the upper chest/throat region of the same animal showing background (grey color) and IRDye800 channel (green color) 3 h post injection of the probe (red arrowhead: cuff location; downward arrowhead: downstream region; upward arrowhead: upstream region; red arrow: LCCA). (**D**) Intensity data from MSOT experiments after the indicated time points post-surgery (*n* = 5–9 for each time point; grey boxes: upstream region; patterned boxes: downstream region; white boxes: LCCA). The upstream part of the RCCA exhibits a significantly enhanced intensity at all recorded time points, while the downstream region does show an intensity comparable to that of the LCCA (* *p* < 0.05, *** *p* < 0.005, ns: not significant).

**Figure 2 ijms-24-08627-f002:**
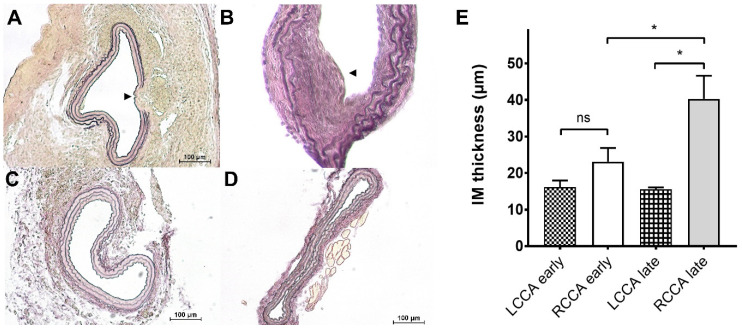
Elastica van Gieson stainings of explanted common carotid arteries. (**A**) RCCA sample from the upstream region 12 weeks post-surgery. A considerable degree of intima impairment can be observed (arrowhead). (**B**) RCCA sample 12 weeks post-surgery from within the cuff also showing signs of endothelial abnormalities (arrowhead). (**C**) RCCA sample from the upstream region 4 weeks post-surgery, where no intimal thickening can be detected. (**D**) The left common carotid artery 12 weeks post-surgery served as control. (**E**) Analysis of the intima media thickness of early (2 and 4 weeks post-surgery) and late (10 and 12 weeks post-surgery) explants of carotid arteries (upstream of the implant). A significantly enhanced thickness is observed in the late tissue samples (*n* = 6–9, one-way ANOVA; * *p* < 0.05, ns: not significant).

**Figure 3 ijms-24-08627-f003:**
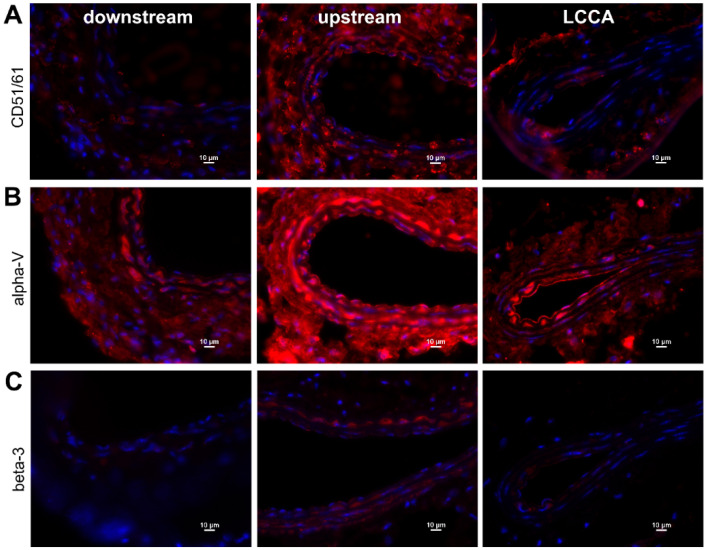
Immunohistochemical identification of integrin α_v_ and β_3_ subunits on (from left to right) sections from the region downstream of the cuff, sections from upstream of the cuff, and LCCA control sections (blue color: DAPI nuclear stain). (**A**) Fluorescence microscopy of CD51/CD61 antibody staining depicting α_v_β_3_ heterodimer. (**B**) Staining for integrin alpha-V subunit. (**C**) Staining for integrin beta-3 subunit. Magnification 40×; illumination 10s (Cy 5) and 1s (DAPI).

**Figure 4 ijms-24-08627-f004:**
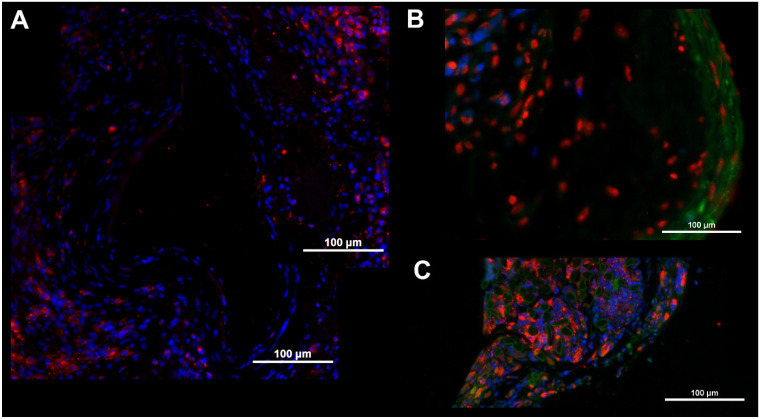
Immunohistochemical identification of macrophages with CD68 antibody. (**A**) Fluorescence microscopy of CD68 antibody staining in a vessel 12 weeks post-surgery (upstream; red color: CD68, blue color: DAPI nuclear stain, stitched image). (**B**) Fluorescence microscopy of CD68 antibody staining in a vessel 4 weeks post-surgery (upstream, green color: SMA). (**C**) Fluorescence microscopy of CD68 antibody staining in a vessel 4 weeks post-surgery, depicting tissue directly surrounding the cuff material (green color: SMA).

**Figure 5 ijms-24-08627-f005:**
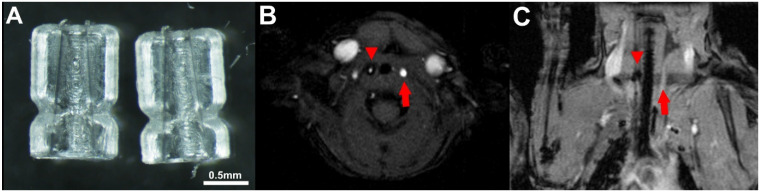
Images showing the cuff material and the localization in vivo and ex vivo. (**A**) Image showing both parts of the tapered cuff before surgery. The scale provides an impression of the dimensions. (**B**,**C**) MRI images as control of the localization after surgery. The red arrowhead points at the cuff around the RCCA, and the red arrow at the control LCCA without cuff.

**Figure 6 ijms-24-08627-f006:**
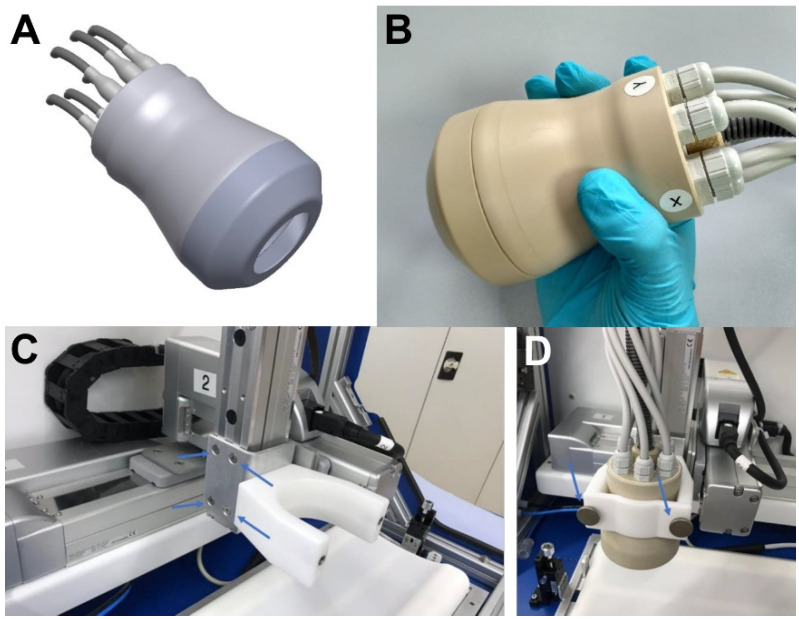
Design and implementation of the hand-held MSOT probe. (**A**) Schematic image showing the design of the probe. (**B**) Application example as hand-held. (**C**) Mounting device for a fixed installation of the probe above the animal bed. (**D**) Image showing the probe in the mounting device ready to use (images courtesy of iThera Medical).

## Data Availability

The original contributions presented in the study are included in the article/Appendix A, further inquiries can be directed to the corresponding author.

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
