# Peer review of "Detection of Early Endothelial Dysfunction by Optoacoustic Tomography"

_ijms, 2023, doi:10.3390/ijms24108627_

Round 1

Reviewer 1 Report

The authors have detailed a procedure aimed at detecting the early stages of vascular complications in vivo through the use of optoacoustic tomography. Briefly, the authors employ a surgical procedure for invoking endothelial dysfunction in a rodent model as an alternative to genetically modified animals that have in recent times become the model of choice but limit the ability to explore the ‘natural’ onset and development of endothelial dysfunction. Utilising a carotid artery cuff approach, the authors were able to track the onset of problem regions in the vascular tree over time by tracking changes in the fluorescent signal emitted at sites where changes in the RGD motif of particular integrins had occurred. In summation, these data suggest optoacoustic tomography may hold promise as an approach to tracking changes in vascular homeostasis, though more work to test its capabilities is warranted.

In reviewing the manuscript I made a number of observations. The following should be considered when preparing a suitable resubmission.

1.       The authors indicate margins in the n-number of animals used. Initially it is stated that animals 8-10 were prepared for the study, and in some data sets the groups presented vary between 5-9. Justification for why there are differences in these n-numbers needs to be provided by the authors.

2.       A mean age of the animals is given – can the SD of this age be provided?

3.       There are instances where the authors allude to previous methods – for example in the surgical procedure. It is this reviewers belief that the protocol be provided to ensure the readership is in receipt of all pertinent details for these data and to ensure repeatability.

4.       The authors do not specify which tissue was utilised specifically for the immunohistochemistry studies.

5.       How were the regions examined for the CIMT study selected? Were these consistent across samples or were these selected at random by a software?

6.       For the imaging studies, it would be useful to know how many animals these images represent. How many animals were these stainings performed on, and were the results consistent?  

7.       Did the authors attempt to perform any kind of quantitation on the staining obtained?

Author Response

Dear Reviewer of manuscript ijms-2391786.

Thank you very much for your valuable comments and suggestions. I tried to address the listed issues to the best of my knowledge. Unfortunately, figure 2 was missing in the template-formatted manuscript file. It showed the hardware used for the acquisition of MSOT data, but no acquired data per se. In the now revised manuscript file, figure 2, including caption, is inserted again on page 5 in section 2.3. If there are any issues with this figure, I will be happy to revise then as demanded.

Addressed issues:

I understand the confusion about animal numbers. Therefore, I corrected the initial animal number (line 101) and explained deviations at the imaging time-point (line 163ff) and the analysis section (line 194ff).

  • The same holds true for the animal age. I explained the numbers in lines 102ff and 110ff (however, I did not calculate standard deviations. We do not receive exact birth dates for every animal we receive).
  • The surgical procedure is now described in more detail (lines 115ff).
  • Also, the harvesting of tissue for further analysis is now described in more detail in lines 198ff.
  • The regions where the CIMT was measured was indeed chosen randomly. This has been included in the description of the method in line 223.
  • The in vivo images shown in figure 3 (B/C) only depict images of one animal from three different angles as supplied by the viewMSOT software. This is now mentioned in the caption to the figure (line 266ff). Immunohistochemical staining were conducted in at least two different sections from the same animal. Every time point post-surgery was examined, however not every histological image was further evaluated. We focused on meaningful and robust images.
  • Also, a statistical analysis of the acquired images was not performed.

Reviewer 2 Report

In the present study the authors wanted to elucidate the applicability of the MSOT hand-held optoacoustic imaging probe in combination with an integrin avb3-targeted fluorescent small molecular RGD mimetic for the detection of early endothelial dysfunction in a cuff model of altered shear stress in carotid arteries of wild-type mice. The novelty of this work is given by the fact that a study of endothelial activation in wild-type animals without additional high-fat diet utilizing the cuff model has not yet been described.

I recommend few minor revisions: 

- please add the licence code of GraphPad Prism used for statistical analysis

- please add the scale bar in all the microscopy images (in figure 2 are missing) 

- I suggest to divide the results in sub-chapters to simplify the comprehension 

- please explain more in details (possibly in the discussion) the importance of this work in the light of its potential application on human beings 

- it has been described that the glycocalyx (sensor of shear stress) is involved in endothelial dysfuncion and plays a role in the interaction with integrins: have you ever considered to analyze also the expression of glycolalyx markers in your experimental setting? 

Minor editing of English language required, especially in the discussion chapter 

Author Response

Dear Reviewer of manuscript ijms-2391786.

Thank you very much for your valuable comments and suggestions. I tried to address the listed issues to the best of my knowledge. Unfortunately, figure 2 was missing in the template-formatted manuscript file. It showed the hardware used for the acquisition of MSOT data, but no acquired data per se. In the now revised manuscript file, figure 2, including caption, is inserted again on page 5 in section 2.3. If there are any issues with this figure, I will be happy to revise then as demanded.

Addressed issues:

  • I cannot access the license code, only in part it is visible in the about tab. However, I expanded the software information in the materials section in line 190ff and 227ff.
  • In figure 3A scale bars have been added. For the MSOT images (B/C) this was not possible.
  • The results section has been subdivided into three sub-chapters.
  • The translational aspect of the described method has been emphasized in the discussion section in lines 424ff.
  • The importance of the glycocalyx for transducing shear stress into cells has not yet been in the focus of our group. However, a very recent review article very wells summarizes the diversity of the possible components acting as mechanotransducers on endothelial cells in the vasculature, including the glycocalyx and integrins. This reference [#35] and a respective note (lines 382ff) have been added to the manuscript.
  • The whole manuscript but especially the discussion section have been proof-read and minor linguistic inconsistencies have been corrected.

Round 2

Reviewer 1 Report

The authors have suitably addressed my comments and the manuscript is much improved.